# Unobtrusive measures of prejudice: Estimating percentages of public beliefs and behaviours

**Adrian Furnham[1], Jan Ketil Arnulf[1]\*, Charlotte Robinson[2]**

1 Department of Leadership and Organisational Behaviour, Norwegian Business School (BI), Olso, Norway,
2 Department of Psychology, University of Bath, Bath, United Kingdom

\* jan.k.arnulf@bi.no

**Data Availability Statement:** All relevant data are within the manuscript and its Supporting Information files.

**Funding:** The authors received no specific funding for this work.

## Abstract

This study was concerned with how accurate people are in their knowledge of population norms and statistics concerning such things as the economic, health and religious status of a nation and how those estimates are related to their own demography (e.g age, sex), ideology (political and religious beliefs) and intelligence. Just over 600 adults were asked to make 25 population estimates for Great Britain, including religious (church/mosque attendance) and economic (income, state benefits, car/house ownership) factors as well as estimates like the number of gay people, immigrants, smokers etc. They were reasonably accurate for things like car ownership, criminal record, vegetarianism and voting but seriously overestimated numbers related to minorities such as the prevalence of gay people, muslims and people not born in the UK. Conversely there was a significant underestimation of people receiving state benefits, having a criminal record or a private health insurance. Correlations between select variables and magnitude and absolute accuracy showed religiousness and IQ most significant correlates. Religious people were less, and intelligent people more, accurate in their estimates. A factor analysis of the estimates revealed five interpretable factors. Regressions were calculated onto these factors and showed how these individual differences accounted for as much as 14% of the variance. Implications and limitations are acknowledged.

## Introduction

*"There are three kinds of lies: lies, damned lies, and statistics"* (Benjamin Disreali); *"Nothing is so fallacious as facts, except figures."* (George Canning); *"Statistics are like bikinis. What they reveal is suggestive, but what they conceal is vital."* (Aaron Levinstein); *"He uses statistics as a drunken man uses lamp-posts, for support rather than for illumination."* (Andrew Lang); *"I never believe in statistics unless I've forged them myself."* (Winston Churchill).

What percentage of the citizens of your country are gay, immigrants, divorced or Muslims? What does the accuracy of this estimate say about your religious and political beliefs? Do people who greatly over- or under-estimate these sorts of national statistics attempt to justify their contentious view? Is the fact that people are misinformed about population norms a major contributory factor to their personal religious and political beliefs? This study sought to answer

**Competing interests:** The authors have declared that no competing interests exist.

some of those questions. It tests the proposition that our perception of statistical facts is always in the service of our psychological mindset, driven by such things as intelligence and defense mechanisms. We test the hypothesis that group belongingness, ideology and age-related predicaments drive tendencies to distort facts while IQ, or ego-strength, has a moderating influence.

## Misperceptions of risk and danger

In a relevant and important book, Rosling et al. [1] suggests that most people are wrong about the state of the world, misbelieving it is poorer, less healthy, and more dangerous than it actually is, and that attributing this not to random chance but to misinformation. They identify ten "instincts" that lead people to make mistakes, called, for instance, the "gap", "straight-line", "destiny", "blame" or "urgency" instinct. They give examples and rules of thumb to overcome this so as not to make factual mistakes. Pertaining to the present study, Rosling found that being misinformed is not a random mistake. Rather, the misinformation is skewed towards perceptions of danger. Thus, people are prone to overestimate numbers that could pose a danger to them (such as the incidence of terrorism), while "imminent dangers" are very much a matter of personal interpretation [2].

This attenuation of possible threats has also been noted by Kahneman and Tversky in their work on biased judgements [3, 4]. A particularly salient feature of biased statistical judgement is what they call "availability heuristic"; scary pieces of information draw more attention and hence change the perception of risk. Adding to that, the "anchoring effect" makes people susceptible to changing the scale of reference points, making an anxiety-provoking number appear more credible and less statistically suspect to the individual. Thus, we have a natural risk-oriented bias in statistical assumptions. Risk does not exist 'out there' because personally relevant risks will be perceived as more looming and imminent than vague, general statistical threats, which will be similarly downplayed or ignored. This shapes the individual psychological dynamics underlying attitudes and demographically determined beliefs.

## Assessment of ideology

Personality and social psychologists have always been interested in socio-political ideology and developed tests to measure political, social and religious attitudes [5]. Hence there are measures of authoritarianism, conservatism, conspiracy theories, dogmatism, ethnocentrism, fascism, Machiavellianism, paranoia, racism, and social dominance. Many tests, and the theories from which they are derived, reflect the issues of the day and their particular time period [6–10]. Many studies have also been concerned with individual difference correlates of these measure [11–14]. There also many studies on various types of prejudice [15, 16].

However, one of the more difficult aspect of measuring any form of ideology, particularly if it is associated with prejudice, are issues of impression formation, dissimulation and faking which inevitably threaten the validity of the measurement. The idea is that respondents easily pick up what researchers are trying to measure and, being sensitive to social mores, do not respond honestly: this has been a consistent finding since the 1930s [17]. In short, they are "politically correct" rather than honest. Most researchers in the area also acknowledge what is called subtle prejudice racism, also called casual racism [18, 19]. One question then is how can prejudicial, indeed all ideological, beliefs be more accurately assessed?

Cognitive psychologists have also been interested in these topics. Bartlett [20] argued that the past is reconstructed to conform with current cognitive states. Schemata theory suggests that people remember more information that is congruent with their attitudes, because the attitudes act as an organizing framework that helps in the encoding and retrieval of attitude-

supportive material. For instance, Furnham and Singh [21] asked male and female adolescents to listen to 15 pro- and 15 anti-female "research findings" concerning sex differences, and later recall details. The results supported the prediction that males and those with more negative attitudes towards women recalled less pro- and more anti-female items, and vice versa.

Taken together, this literature review indicates that several areas of psychology have contributed to a body of literature documenting and predicting that people will be biased in their recall of statistics. Several publications have listed and looked into a range of such biases, where one of the final and most relevant to our work may be confirmation bias, which connotes the seeking or interpreting of often statistical evidence in ways that are partial to existing beliefs, expectations, or a hypothesis in hand [4, 22, 23]. Thus, we know people tend to underestimate risk when they make decisions based on their own experience with rare events, while the overestimation of rare events can happen based on heavy media coverage [24]. Another is the growing literature on conspiracy theories where people select and distort a range of statistics to fit in with their word view [11], probably being successful by stimulating the mechanisms reviewed above.

## This study

This is an exploratory study concerning people's population estimates of a variety of socio-economic and other issues. It seems the case that politicians and lay people frequently quote statistics about population norms to justify their position on various issues. Sometimes these statistics are challenged, but often not as it takes some effort to find the actual data. Hence people might disagree on many issues, particularly issues of social class, race and religion. They quote statistics that suit their argument, whether they know them to be true or not. Moreover, they may recall much better statistics which suit their particular position.

There are indeed popular books and media programmes and blogs on peoples' misunderstanding and misuse of statistics [25]. This study is however more about their knowledge of population statistics and the correlates of that knowledge.

This study examined two things: on what issues people are more or less accurate, and what the individual difference correlates of these estimates are. We concentrated on demography (sex, age, education), ideology (religious and political beliefs) and intelligence. There are various studies linking intelligence levels to prejudice. It was our hypothesis that religious and political beliefs would be most associated with estimates, in predictable ways. That is, more politically right-wing people would over-estimate the number of migrants and those on state benefits, while more religious people would over-estimate those visiting places of worship. Also, that where education and intelligence correlated significantly, those estimates tended to be more accurate. However, this is in many ways an exploratory study.

## Method

### Participants

A total of 616 participants completed the questionnaire: 307 were men and 309 were women. They ranged in age from 18 to 70 years, with a mean age of 31.1 years ($SD$ = 9.54 years). All had secondary school education and just over half were graduates. In total, 46.2% were single and 26.8% married, and 69% had no children. They rated themselves on three scales: How religious are you? (Not all at = 1 to Very = 8) ($M$ = 2.97, $SD$ = 2.42); How would you describe your political beliefs? (Very Left Wing = 1 to Very Right Wing = 8) ($M$ = 6.07, $SD$ = 1.63); How optimistic are you? (Not at all = 1 to Very = 8) ($M$ = 4.70, $SD$ = 2.17). In all, 36.2% said they believed in the after-life and 63.7% said they did not.

## Measures

1. *Knowing Britain*: A questionnaire was devised for use in this study. It was titled "What do you know about Britain". There were 25 questions which can be seen in Table 1. They were chosen to cover a wide range of topics that interest demographers and where we had data so that we could verify the accuracy of estimates. Each question requested an open-answer as opposed to ticking pre-set categories.

2. *Wonderlic Personnel Test* [26]. This 50-item test can be administered in 12 minutes and measures general intelligence. Items include word and number comparisons, disarranged sentences, story problems that require mathematical and logical solutions. It is a short measure of fluid intelligence. The test has impressive norms and correlates very highly (r = .92) with the WAIS-R. In this study, we used 16 items from Form A (14, 15, 18, 21, 24, 27, 28, 29, 30, 32, 33, 34, 36, 37, 43, 46). The measure has been used in many studies [27, 28].

## Procedure

Departmental ethical approval was gained prior to data collection (CEHP/514/2017). Data was collected on-line through *Prolific*, a platform like the better-known Amazon-Turk. It was conducted in early 2021. The participants were predominantly British. We would expect British citizens to be better informed than people with little or no knowledge of the country.

**Table 1. Mean estimates for the twenty questions.**

| Item | Mean | SD |
|---|---|---|
| What is the average UK annual salary per year (in £)? | 30254.55 | 18728.93 |
| What percentage of the adult British population are Muslims? | 14.11 | 10.92 |
| What percentage of the adult British population are vegetarians? | 14.30 | 10.86 |
| What percentage of the adult British population are gay? | 14.48 | 12.66 |
| What percentage of the adult British population were not born in the UK? | 22.08 | 15.11 |
| What percentage of the adult British population visit a church once a week? | 20.72 | 15.79 |
| What percentage of the adult British population own their own home? | 42.57 | 18.49 |
| What percentage of the adult British population own a car? | 62.99 | 18.09 |
| What percentage of the adult British population are left-handed? | 23.96 | 14.76 |
| What percentage of the adult British population are smokers? | 33.43 | 16.89 |
| What percentage of the adult British population voted in the last election? | 61.45 | 13.69 |
| What percentage of the adult British population visit a mosque once a week? | 12.87 | 12.36 |
| What percentage of the adult British population send their children to private schools? | 19.02 | 14.36 |
| What percentage of the adult British population own more than one house | 13.87 | 11.59 |
| What percentage of the adult British population claim state benefits? | 29.12 | 18.35 |
| What percentage of the adult British population earn over £50,000 a year? | 24.09 | 15.91 |
| What percentage of the adult British population have private health insurance? | 30.31 | 20.61 |
| What percentage of the adult British population have a criminal record? | 14.92 | 13.38 |
| What percentage of the adult British population are over 65 and retired? | 29.78 | 14.79 |
| What percentage of the adult British population have a dual nationality? | 19.91 | 15.18 |
| What percentage of the adult British population have a registered disability? | 13.72 | 10.33 |
| What percentage of the adult British population are university graduates? | 37.04 | 17.24 |
| What percentage of the adult British population get divorced? | 34.74 | 16.56 |
| What percentage of the adult British population die of cancer? | 20.48 | 15.93 |
| What percentage of the adult British population live in London? | 19.74 | 14.56 |

Participants were compensated for their time (receiving £1.75). Usual data cleansing and checking led to around 2% of the 630 recruited being rejected before further analysis. The study was run in March 2021.

## Results

A first inspection was done for missing data. Two things were apparent: first, there was around 7% data missing, where participants had chosen not to answer any of the questions. Second, there was very little missing data in the sense that some questions were answered and not others. Thus, the analysis was performed on an N = 573.

Table 1 shows the mean estimates for the 25 items, while Table 2 shows the actual data available. The first observation is that there is a general tendency to over-estimate nearly all

**Table 2. Actual data.**

| Item | Real % | Reference |
|---|---|---|
| 1. What is the average UK annual salary per year (in £)? | £38,600 | ONS (2020) via website https://bit.ly/38OS2ns |
| 2. What percentage of the adult British population are Muslims? | 5.1% | ONS (2018) https://bit.ly/2OEPaTx |
| 3. What percentage are vegetarians? | 14% | Finder.com (2020) https://bit.ly/3bNmY9H |
| 4. What percentage are gay? | 2.2% | ONS (2018) https://bit.ly/3bR5olj |
| 5. What percentage were not born in the UK? | 13.6% | ONS (2020) https://bit.ly/2Q0ys14 |
| 6. What percentage visit a church once a week? | 1.28% | Church Times (2019) https://bit.ly/3cB48Sj |
| 7. What percentage own their own home? | 63% | GOV.UK (2018) https://bit.ly/38Ktf4h |
| 8. What percentage own a car? | 66% | Statista (2018) https://bit.ly/3lkcodz |
| 9. What percentage are left-handed? | 10% | Healthline (2019) https://bit.ly/3cz9qOl |
| 10. What percentage are smokers? | 14.1% | ONS (2019) https://bit.ly/30PKcpj |
| 11. What percentage voted in the last election? | 67.3% | Parliament (2019) https://bit.ly/3qTVtzD |
| 12. What percentage visit a mosque once a week? | 1% | Express 2020 estimate* https://bit.ly/3cCeMZ9 |
| 13. What percentage send their children to private schools? | 7% | GOV.UK (2019) https://bit.ly/3r0yD9L |
| 14. What percentage own more than one house? | 10% | City AM (2019) https://bit.ly/3rRgPik |
| 15. What percentage claim state benefits? | 53% | GOV.UK (2019) https://bit.ly/3toOfFl |
| 16. What percentage earn over £50,000 a year? | 10% | Student Room & BBC articles (2019) https://bit.ly/2Q5PWsZ https://bbc.in/2QbgUzy |
| 17. What percentage have private health insurance? | 11% | KingsFund (2014) https://bit.ly/3thDeWu |
| 18. What percentage have a criminal record? | 17% | Home Office (2017) https://bit.ly/3rRYL7R |
| 19. What percentage are over 65 and retired? | 18.5% | ONS (2019) https://bit.ly/3czCntz |
| 20. What percentage have a dual nationality? | 1% | International Advisor (2020) https://bit.ly/3lk7P2I |
| 21. What percentage have a registered disability? | 18% | St Andrews (?) https://bit.ly/3eIYf8l |
| 22. What percentage are university graduates? | 21.2% | ONS (2017) https://bit.ly/30LZITp |
| 23. What percentage get divorced? | 42% | Crisp & Co (2019) https://bit.ly/3eKEs8D |
| 24. What percentage die of cancer? | 0.3% | Macmillan (2016) https://bit.ly/3bRDdTm |
| 25. What percentage live in London? | 13.7% | Statista (2019) https://bit.ly/3vyU7xP |

items. Those that showed most accuracy were estimates of people who were vegetarian, car owners and voted in the last election, while those items which showed overestimates for more than 10% included people who were gay (12%). There were a few items which showed a significant under-estimate, including those who claim state benefits. Table 1 also illustrates the size of the standard deviations which suggested little agreement on many of these issues.

As an initial inspection of the data, three demographic factors (sex, age, education) and three ideological factors (religion, politics, life-after-death) were correlated with the 25 estimates. A clear pattern emerged: few of the correlations (at p < .01) with demography were significant (sex: 6 were significant; age: 8 were significant; education: 0 were significant) while there were many for the ideological factors (religion:14 were significant; political beliefs: 7 were significant; life-after-death: 12 were significant). The estimates which showed some of the highest correlations included issues concerned with religion (number of Muslims, mosque attendance) as well as socio-economic issues (19: private schools; home ownership).

All respondents noted their nationality as free text and the responses contained a total of 95 nationalities. These were not always mutually exclusive (some respondents listing up to 3 nationalities, others entering ethnic subgroups in countries), but the majority seemed to hold non-UK citizenships. Assuming that growing up in Britain might allow more accurate local knowledge, we computed a dichotomous version of this variable, assigning the value 1 to everyone who listed "British" as their first and only entry and 0 to all others. Based on this procedure, 71 persons (11.5%) were categorized as exclusively "British". We want to emphasize that no other assumptions were made about this variable than making a crude distinction between obviously localized and possibly late acquired knowledge about the British society.

The general picture is that people do not seem very accurate in their "guestimates", as the average responses to all questions are significantly different from the true numbers except for question 3, the number of vegetarians. Of the remaining 24 questions, people tend to overestimate 16 of them while only 8 tend to be underestimates, see Table 3. The same table also shows two the correlations between the 25 questions and the three attitudes, religiousness, political orientation and optimism, and IQ. In general, people with a strong religious belief tend to overestimate, while a conservative political orientation and more importantly, IQ, seems related to more realistic estimates. Optimism does not seem to bear any particular relationship with the opinions on these issues.

Table 3 shows the correlations between two numbers relating to the estimates and four individual difference variables. It distinguishes between two different aspects of the responses: The "magnitude" of a response simply represents the raw score on that variable while the "accuracy" reflects how close the respondent comes to the correct value, in absolute numbers. This makes an interesting difference because the accuracy in itself may simply represent knowledge about an issue, or the lack of it. In this case, being off target on the lower side is equal to be off target on the higher. The magnitude, however, reflects a tendency to exaggerate, a mind-set in addition to being uninformed. As Table 3 shows, the tendency to exaggerate is more strongly related to attitudes than the mere accuracy. However, IQ generally works in the opposite direction. Note that "inaccuracy" is measured in percentage deviation *away from* the accurate number (taking the absolute number of 100% correct estimate minus the percentage of deviation). Therefore, correlations with inaccuracy are expected negative when getting closer to the real census statistic.

Since there is a clear tendency to overestimate a major subset of the questions, and since the exaggerations seem more strongly related to our variables of interest, we decided to explore the data for patterns in "guestimates". Subjecting the 25 raw estimates variables to a principal component analysis (PCA) with varimax rotation, the screen plot indicated four factors, see Table 4. The four factors explained together 44.21% of the variation and their Eigenvalues

**Table 3. Correlations between responses to the 25 questions, three attitudes and IQ–by magnitude and absolute inaccuracy.**

| Question | | Religiousness | Political | Optimism | IQ |
|---|---|---|---|---|---|
| Average UK annual salary per year (in £)? | Magnitude | .06 | .00 | .00 | .01 |
| *(Significant underestimate)* | InAccuracy | .10* | .00 | .02 | -.18** |
| % of British population are Muslims? | Magnitude | .14** | .00 | .00 | -.23** |
| **(Significant overestimate)** | InAccuracy | .14** | .00 | .00 | -.23** |
| % are vegetarians? | Magnitude | .20** | -.09* | .01 | -.24** |
| ***Accurate*** | InAccuracy | .11* | -.09* | .05 | -.16** |
| % are gay? | Magnitude | .04 | .04 | .00 | -.18** |
| **(Significant overestimate)** | InAccuracy | .04 | .04 | .00 | -.18** |
| % not born in the UK? | Magnitude | .10* | -.11** | .00 | -.20** |
| **(Significant overestimate)** | InAccuracy | .07 | -.10* | .00 | -.18** |
| % visit a church once a week? | Magnitude | .15** | .00 | .04 | .00 |
| **(Significant overestimate)** | InAccuracy | .15** | .00 | .04 | .00 |
| % own their own home? | Magnitude | .11* | -.12** | .00 | .00 |
| *(Significant underestimate)* | InAccuracy | .00 | .10* | .01 | .00 |
| % own a car? | Magnitude | .07 | -.12** | .00 | .00 |
| *(Significant underestimate)* | InAccuracy | .01 | .02 | .01 | .00 |
| % are left-handed? | Magnitude | .09* | .00 | .00 | .00 |
| **(Significant overestimate)** | InAccuracy | .09* | .00 | .00 | .00 |
| % are smokers? | Magnitude | .04 | -.09* | .03 | -.11** |
| **(Significant overestimate)** | InAccuracy | .05 | -.09* | .04 | -.12** |
| % voted in the last election? | Magnitude | .11* | .00 | .02 | .00 |
| *(Significant underestimate)* | InAccuracy | .00 | .02 | .00 | -.11** |
| % visit a mosque once a week? | Magnitude | .29** | -.13** | .00 | -.30** |
| **(Significant overestimate)** | InAccuracy | .29** | -.13** | .00 | -.30** |
| % send their children to private schools? | Magnitude | .17** | .00 | .01 | -.19** |
| **(Significant overestimate)** | InAccuracy | .16** | .00 | .01 | -.19** |
| % own more than one house? | Magnitude | .22** | -.19** | .00 | -.16** |
| **(Significant overestimate)** | InAccuracy | .18** | -.19** | .00 | -.17** |
| % claim state benefits? | Magnitude | .11** | -.20** | .01 | -.19** |
| *(Significant underestimate)* | InAccuracy | -.10* | .18** | .00 | .17** |
| % earn over £50,000 a year? | Magnitude | .13** | .00 | .00 | .00 |
| **(Significant overestimate)** | | | | | .00 |
| % have private health insurance? | Magnitude | .09* | -.10* | .00 | -.10* |
| **(Significant overestimate)** | InAccuracy | .08 | -.09* | .00 | -.10** |
| % have a criminal record? | Magnitude | .12** | -.10* | .04 | -.21** |
| *(Significant underestimate)* | InAccuracy | .06 | .00 | .07 | -.15** |
| % are over 65 and retired? | Magnitude | .15** | .00 | .00 | -.15** |
| **(Significant overestimate)** | InAccuracy | .16** | .00 | .00 | -.15** |
| % have a dual nationality? | Magnitude | .14** | .00 | .00 | -.20** |
| **(Significant overestimate)** | InAccuracy | .14** | .00 | .00 | -.20** |
| % have a registered disability? | Magnitude | .11** | .00 | .00 | -.16** |
| *(Significant underestimate)* | InAccuracy | .00 | .02 | .00 | .03 |
| % are university graduates? | Magnitude | .15** | .00 | .02 | .00 |
| **(Significant overestimate)** | InAccuracy | .15** | .00 | .04 | .00 |
| % get divorced? | Magnitude | .03 | .00 | .06 | -.10** |
| *(Significant underestimate)* | InAccuracy | .07 | .04 | .00 | .00 |
| % die of cancer? | Magnitude | .06 | -.09* | .01 | -.16** |

*(Continued)*

**Table 3.** (Continued)

| Question | | Religiousness | Political | Optimism | IQ |
|---|---|---|---|---|---|
| **(Significant overestimate)** | InAccuracy | .06 | -.09* | .01 | -.16** |
| % live in London? | Magnitude | .23** | -.14** | .00 | -.26** |
| **(Significant overestimate)** | InAccuracy | .19** | -.12** | .03 | -.25** |

(*Note that "accuracy" is calculated as absolute deviance from the correct number in percent. A higher correlation reflects a tendency to overestimate, or, in the case of accuracy, to be off target. Negative correlations with accuracy reflects a tendency towards more accurate guestimates.)

spanned 6.15–1.45. The four factors seem to make sense, and we label them "Minorities" (not born in the UK, Muslims, gays, left-handed, etc.), "Welfare" (divorce, retirement, cancer), "Affluence" (owning property and cars), and "Education" (university degree and income).

We then regressed our variables of interest–religiousness, political orientation, optimism, and IQ–on the four factors, as well as the average inaccuracy. We used demographics as control variables and the results are displayed in Table 5. It appears that the ideological variables contribute most to exaggerated estimates of minorities, and second most to matters concerning welfare. The tendency to exaggerate these matters are tempered by IQ, which contributes

**Table 4. Rotated component matrix of the 25 item responses.**

| | Component | | | |
|---|---|---|---|---|
| | 1 | 2 | 3 | 4 |
| % MUSLIMS | .697 | | | |
| % UKBORN | .623 | | | |
| % VEGET | 620 | | | |
| % DUALNAT | .564 | .395 | | |
| % GAY | .557 | .399 | | |
| % SMOKER | .542 | | | |
| % MOSQUES | .531 | | | |
| % CHURCH | .508 | | | |
| % LEFTHAND | .507 | | | |
| % PRIVSCHOOL | .505 | | | .352 |
| % CANCER | | .688 | | |
| % DISABILITY | | .637 | | |
| % DIVORCFE | | .565 | | |
| % RETIRED | | .497 | | |
| % LONDON | | .449 | | |
| % CAR | | | .738 | |
| % UKHOME | | | .640 | |
| % BENEFTS | | | .585 | |
| % CRIM | | .428 | .548 | |
| % HOUSEOWN | | | .541 | |
| % EARN | | | | .755 |
| % HEALTH | | | | .641 |
| % SALARY | | | | .440 |
| % UNI | | | | .421 |
| % VOTE | | | | |

(Explained variance 44.21%, Eigenvalues 6.15–1.45).

**Table 5. Regressions onto four factors and average inaccuracy.**

| | Minorities | | | | Welfare | | | | Affluence | | | | Education | | | | Inaccuracy Average | | | |
|---|---|---|---|---|---|---|---|---|---|---|---|---|---|---|---|---|---|---|---|---|
| | B | SE | Beta | t | B | SE | Beta | t | B | SE | Beta | t | B | SE | Beta | t | B | SE | Beta | t |
| Sex | .26 | .09 | .14** | 2.91 | .43 | .09 | .22** | 4.77 | -.06 | .10 | -.03 | -0.66 | .04 | .10 | .02 | 0.45 | .91 | .27 | .15** | 3.37 |
| Birth Year | .01 | .01 | .13* | 2.32 | .01 | .01 | .12* | 2.19 | .02 | .01 | .14* | 2.52 | .01 | .01 | .06 | 1.11 | .03 | .02 | .08 | 1.65 |
| Schooling | -.00 | .01 | -.01 | -0.11 | -.01 | .01 | -.02 | -0.43 | .00 | .01 | .01 | 0.14 | .01 | .01 | .04 | 0.80 | .02 | .04 | .02 | 0.38 |
| Degrees | .04 | .10 | .02 | 0.45 | .13 | .10 | .64 | 1.32 | .22 | .11 | .10* | 2.03 | -.06 | .11 | -.03 | -0.58 | .72 | .30 | .11 | 2.45 |
| Married | .01 | .03 | .02 | 0.40 | .02 | .03 | .03 | 0.54 | -.03 | .03 | -.05 | -0.98 | .00 | .03 | .00 | -0.00 | .16 | .08 | .08 | 1.90 |
| British | .03 | .14 | .01 | 0.22 | .57 | .15 | .19** | 3.95 | .00 | .16 | .00 | 0.02 | -.75 | .16 | -.25** | -4.86 | 1.62 | .43 | .17** | 3.74 |
| Children | -.13 | .05 | -.13* | -2.41 | -.03 | .06 | -.03 | -0.57 | .04 | .06 | .04 | 0.68 | .02 | .06 | .02 | 0.27 | -.41 | .17 | -.13** | -2.46 |
| Occup. | .05 | .04 | .06 | 1.20 | .07 | .04 | .08 | 1.77 | .05 | .43 | .05 | 1.04 | .08 | .04 | .09 | 1.82 | .23 | .12 | .08* | 1.91 |
| Religious | .07 | .02 | .17** | 3.58 | .04 | .02 | .09 | 1.90 | .05 | .02 | .11* | 2.27 | .04 | .02 | .10* | 2.05 | .18 | .06 | .14** | 3.07 |
| Politics | -.04 | .03 | -.07 | -1.50 | -.06 | .03 | -.10* | -1.99 | -.09 | .03 | -.15** | -2.99 | -.04 | .03 | -.07 | -1.32 | -.17 | .08 | -.10* | -2.05 |
| Optimist | .01 | .02 | .03 | 0.69 | -.01 | .02 | -.02 | -0.39 | .03 | .02 | .06 | 1.19 | -.00 | .02 | -.01 | -0.18 | -.04 | .06 | -.03 | -0.65 |
| IQtotal | -.08 | .02 | -.23** | -4.99 | -.06 | .02 | -.18** | -3.75 | -.03 | .02 | -.08 | -1.60 | -.01 | .02 | -.03 | -0.64 | -.14 | .05 | -.14** | -3.05 |
| Adjusted $R^2$ | .14 | | | | .14 | | | | .08 | | | | .08 | | | | .12 | | | |
| F | 7.31 | | | | 7.34 | | | | 4.23 | | | | 4.47 | | | | 6.80 | | | |
| p | .000 | | | | .000 | | | | .000 | | | | .004 | | | | .000 | | | |

significantly in the opposite direction, while optimism does not seem to play any role. The respondents' perceptions of matters related to affluence and education are much less influenced by ideology.

Younger females with fewer children and who were more intelligent and less religious were more accurate about minorities. Similarly, younger British females who were more left wing and more intelligent were more accurate about welfare. Considering the total accuracy score there were seven significant individual differences predictors in order of magnitude: nationality, sex, religiousness, IQ, number of children, occupation and political views. In all they accounted for 12% of the total variance.

## Discussion

This exploratory study demonstrated a significant relationship between participants demography, ideology and intelligence and their knowledge of a range of demographic statistics. Despite being "bombarded" with statistics by the media, people appear to be ignorant of a range of statistics pertaining to important features of society. Overall, they tended to over-estimate percentages, particularly when the incidence in the population was low such as in the case of going to places of worship and some economic variables (e.g. have private health insurance).

In this study we were particularly interested in individual correlates of the estimates and the accuracy of those estimates. We examined both the actual estimate (magnitude) and the accuracy of those estimates. As noted by Rosling [1], the inaccuracy does not seem to be random, but is patterned by various mindsets in the respondents.

The factor analysis revealed four interpretable factors, though it would be desirable to replicate this. They concerned issues around minority groups of all sorts, and socio-economic factors. Using these as criterion variables our regression showed that some subsets of social statistics are more prone to individual bias than others. Perceptions of education and income-related statistics seem merely inaccurate in that most people agree with no particular individual biases. However, matters pertaining to welfare benefits and minorities seem much more

prone to biases related to religious or political ideological foundations. A possible interpretation of this is that such beliefs are related to the respondents' subjective feeling of social competition and threats. Active religious and political engagement may stem from, or render people sensitive to, signals related to feelings of having their value systems threatened. In such cases, the perceptions of possible threats take on exaggerated proportions reflecting the heightened attention being aroused.

The defensive nature of these exaggerated perceptions is underscored by the way that IQ seems to temper and reduce the estimations. A higher IQ predicts more realistic estimates of social statistics–but only on the features of society that seem to elicit defensive ideological reactions. There are other types of information about society that simply seem to be a matter of local information, and where neither ideology nor IQ plays a big role. In such cases, identifying as "British" does not seem to necessarily imply more accurate estimates, but possibly another source of concern for social statistics. We suggest that IQ is related to education and information gathering (i.e. more depth reading) and hence more knowledge about the topics tested in this study. However, we accept the fact that there are other factors involved with regard to this knowledge such as personal interests (e.g. in current affairs, economics) which may play as important as role as intelligence or education.

One of the first and obvious critiques of a study such as this concerns the reliability of the data shown in Table 2: namely the "objective", "census-type" data. There are those who would dispute and contest many of these statistics as being unreliable, "guestimates" like the number of (openly or not) gay people or those who are (really and consistently) vegetarian. On the other hand, there is probably enough good government collected data on such things as income, nationality and state benefits to suggest that data available is both reasonable, accurate and reliable. In this sense it is difficult to talk about accuracy and radical over- and under-estimates. Inevitably, some people are better informed that others about some of the estimates used here. It seems the case that often in ideological debates about a variety of issues statistical "facts" are bandied about as a way of justifying positions.

There is an extensive literature how people access, store and recall information about all issues including the sort of data we collected in this study. The research suggests that people are highly selective in what they read and recall, and that it is related to pre-existing beliefs. Thus their memory and knowledge, which is relatively easily assessed, may be a good unobtrusive measure of their abilities and beliefs which are more difficult to measure directly. We believe this a fruitful, relatively unexplored, area of research that deserves more attention.

Like all studies this had its limitations. Our sample was not representative of the British population, being on average both younger and better educated. Whilst we had some data on each participant, it would have been desirable to know more about their personal circumstances (e.g., personal income, job history), their community involvement in religion and politics, as well as their confidence in their estimates. All of our measures were short and it would have been preferable to understand more about their political and religious beliefs and behaviours. It would have been desirable to measure their confidence in each of these estimates and use that to weight the items. It should also be pointed out that the intelligence test was not taken under timed conditions. Nevertheless, it is worth pursuing research in this neglected field using statistical estimates as measures of ideology.

## Supporting information

**S1 Data.**
(SAV)

## Author Contributions

**Conceptualization:** Adrian Furnham.

**Data curation:** Adrian Furnham, Charlotte Robinson.

**Formal analysis:** Jan Ketil Arnulf.

**Methodology:** Adrian Furnham.

**Project administration:** Charlotte Robinson.

**Validation:** Charlotte Robinson.

**Writing – original draft:** Adrian Furnham, Jan Ketil Arnulf.

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
