## [Decision Letter · Decision Letter 0]

11 Oct 2021

PONE-D-21-21013Unobtrusive Measures of Prejudice: Estimating percentages of public beliefs and behaviours.PLOS ONE

Dear Dr. Jan Ketil Arnulf,

Thank you for submitting your manuscript to PLOS ONE. After careful consideration, we feel that it has merit but does not fully meet PLOS ONE’s publication criteria as it currently stands. Therefore, we invite you to submit a revised version of the manuscript that addresses the points raised during the review process.

We look forward to receiving your revised manuscript.

Kind regards,

Shah Md Atiqul Haq

Academic Editor

PLOS ONE

2. Please improve statistical reporting and refer to p-values as "p<.001" instead of "p=.000". Our statistical reporting guidelines are available at https://journals.plos.org/plosone/s/submission-guidelines#loc-statistical-reporting

Additional Editor Comments (if provided):

Dear authors,

I would like to revise the paper by following the reviwers' comments and suggestions.

Best regards,

Reviewers' comments:

Reviewer's Responses to Questions

**Comments to the Author**

1. Is the manuscript technically sound, and do the data support the conclusions?

Reviewer #1: Partly

Reviewer #2: Partly

2. Has the statistical analysis been performed appropriately and rigorously? 

Reviewer #1: Yes

Reviewer #2: No

3. Have the authors made all data underlying the findings in their manuscript fully available?

Reviewer #1: Yes

Reviewer #2: No

4. Is the manuscript presented in an intelligible fashion and written in standard English?

Reviewer #1: Yes

Reviewer #2: Yes

5. Review Comments to the Author

Reviewer #1: This article has an interesting topic. It has tried to show that people do not have accurate information about many economic and social facts of their society. Of course, in the introduction of the article, this issue is assumed. So the question is, what was the need for this research?

Of course, in my opinion, this research is necessary because we do not know which characteristics of society are wrong. That is, people's knowledge is far from reality.

This paper in terms of statistics is not complex and the method of work done is not so complicated. The positive point is that the research procedure is clearly described.

I recommend the following tips to improve the quality of the article:

1. Explain the details of the survey in more detail. How many questions were there in the questionnaire? What tools have been used to measure intelligence? Were all the questions open-answer? When was the survey conducted? How do you know if the respondents were British citizens?

2- I think the authors can discuss at the end of the article a little about their understanding of the distance between public opinion and reality and express their theoretical ideas.

3- That smarter people have more accurate information. It seems to be more controversial about this finding. PLease speculate more about this.

Reviewer #2: The goal of the study was to investigate prejudices about socio-demographic situation in Great Britain from a perspective of individual characteristics.

Here, I provide my comments:

1. I propose to rewrite an abstract: include some introduction, goal of the paper.

2. It is not clear based on what literature authors formulated their hypotheses.

3. I recommend to say about nationality of your respondents in the section "Participants". Also, it seems vague to define citizenship based on nationality... How does it cause the results?

4. I did not get how you found correlations in Table 3: could you please provide some example?

5. Page 8: "Three demographic factors (sex, age, education) and three ideological factors (religion, politics, life-after-death) were correlated with the 25 estimates. " - what is the Table for this sentence?

When you put the information about significance of the correlations: it is not clear what are the numbers in breaks (for example, "(sex: 6; age: 8; education: 0)")

6. When you found four new variables in PCA and then used them in regression you did not explain what exactly you used: accuracy of prejudices?

7. Page 11: "five factors were significant predictors of accuracy" - could you please explain this conclusion? It is not clear

+ some additional remarks:

1. page 6: "There are indeed popular books and media programmes and blogs on peoples’ misunderstanding and misuse of statistics (Harford, 2020)." - why did you put this sentence there?

2. page 7: "The measure has been used in many studies (Bertsch & Pesta, 2009)." - but you mentioned only one.

3. I recommend to put together Table 1 and Table 2.

Finally, I recommend publishing this manuscript after major revision (in case authors will be capable to answer all the questions and rearrange the manuscript).

6. PLOS authors have the option to publish the peer review history of their article (what does this mean?). If published, this will include your full peer review and any attached files.

Reviewer #1: **Yes: **Ahmadreza Asgharpourmasouleh

Reviewer #2: No

---

## [Author Response · Author response to Decision Letter 0]

20 Oct 2021

PONE-D-21-21013

Unobtrusive Measures of Prejudice: Estimating percentages of public beliefs and behaviours.

Dear Dr. Haq

Thank you for submitting your email and review of our paper submitted to PLOS ONE. See below how we have reacted to each of the reviewers comments

1. Is the manuscript technically sound, and do the data support the conclusions?

Reviewer #1: Partly

Reviewer #2: Partly

2. Has the statistical analysis been performed appropriately and rigorously? 

Reviewer #1: Yes

Reviewer #2: No

3. Have the authors made all data underlying the findings in their manuscript fully available?

Reviewer #1: Yes

Reviewer #2: No

4. Is the manuscript presented in an intelligible fashion and written in standard English?

Reviewer #1: Yes

Reviewer #2: Yes

5. Review Comments to the Author

)

Reviewer #1: 

This article has an interesting topic. It has tried to show that people do not have accurate information about many economic and social facts of their society. Of course, in the introduction of the article, this issue is assumed. So the question is, what was the need for this research?

Of course, in my opinion, this research is necessary because we do not know which characteristics of society are wrong. That is, people's knowledge is far from reality.

This paper in terms of statistics is not complex and the method of work done is not so complicated. The positive point is that the research procedure is clearly described.

Great News: many thanks

I recommend the following tips to improve the quality of the article:

1. Explain the details of the survey in more detail. How many questions were there in the questionnaire? What tools have been used to measure intelligence? Were all the questions open-answer? When was the survey conducted? How do you know if the respondents were British citizens?

We have answered all these questions in the revision and provided answers to each question.

2- I think the authors can discuss at the end of the article a little about their understanding of the distance between public opinion and reality and express their theoretical ideas.

We have been happy to do this and think it a good suggestion.

3- That smarter people have more accurate information. It seems to be more controversial about this finding. PLease speculate more about this.

We do not find it controversial but have happily speculated on why we found this result.

Reviewer #2: 

The goal of the study was to investigate prejudices about socio-demographic situation in Great Britain from a perspective of individual characteristics.

Here, I provide my comments:

1. I propose to rewrite an abstract: include some introduction, goal of the paper.

OK...done as suggested.

2. It is not clear based on what literature authors formulated their hypotheses.

We have tried to explain this, but given that there is relatively little literature in this precise area we have noted where our hypotheses were speculative

3. I recommend to say about nationality of your respondents in the section "Participants". Also, it seems vague to define citizenship based on nationality... How does it cause the results?

See above...we have added these details and indeed speculated on how the population sample may impact the results

4. I did not get how you found correlations in Table 3: could you please provide some example?

Yes indeed:done as requested

5. Page 8: "Three demographic factors (sex, age, education) and three ideological factors (religion, politics, life-after-death) were correlated with the 25 estimates. " - what is the Table for this sentence?

We have clarified this point

When you put the information about significance of the correlations: it is not clear what are the numbers in breaks (for example, "(sex: 6; age: 8; education: 0)")

Yes: a good observation: we have now explained this

6. When you found four new variables in PCA and then used them in regression you did not explain what exactly you used: accuracy of prejudices?

We have now explained this point

7. Page 11: "five factors were significant predictors of accuracy" - could you please explain this conclusion? It is not clear

Ok....we have attempted to make this much clearer

+ some additional remarks:

1. page 6: "There are indeed popular books and media programmes and blogs on peoples’ misunderstanding and misuse of statistics (Harford, 2020)." - why did you put this sentence there?

Essentially to point out that people are very interested 

2. page 7: "The measure has been used in many studies (Bertsch & Pesta, 2009)." - but you mentioned only one.

We have now referenced other papers.

3. I recommend to put together Table 1 and Table 2.

We have decided not to do this as we believe it is clearer to present the results as we have done initially

Finally, I recommend publishing this manuscript after major revision (in case authors will be capable to answer all the questions and rearrange the manuscript).

Thank you for your suggestions which were most helpful

---

## [Decision Letter · Decision Letter 1]

2 Nov 2021

Unobtrusive Measures of Prejudice: Estimating percentages of public beliefs and behaviours.

PONE-D-21-21013R1

Dear Dr. Jan Ketil Arnulf,

We’re pleased to inform you that your manuscript has been judged scientifically suitable for publication and will be formally accepted for publication once it meets all outstanding technical requirements.

Kind regards,

Shah Md Atiqul Haq

Academic Editor

PLOS ONE

Additional Editor Comments (optional):

Dear authors,

The paper is accepted now.

Congratulations!!

Reviewers' comments:

Reviewer's Responses to Questions

**Comments to the Author**

1. If the authors have adequately addressed your comments raised in a previous round of review and you feel that this manuscript is now acceptable for publication, you may indicate that here to bypass the “Comments to the Author” section, enter your conflict of interest statement in the “Confidential to Editor” section, and submit your "Accept" recommendation.

Reviewer #1: All comments have been addressed

Reviewer #2: All comments have been addressed

2. Is the manuscript technically sound, and do the data support the conclusions?

Reviewer #1: Yes

Reviewer #2: Yes

3. Has the statistical analysis been performed appropriately and rigorously? 

Reviewer #1: Yes

Reviewer #2: Yes

4. Have the authors made all data underlying the findings in their manuscript fully available?

Reviewer #1: Yes

Reviewer #2: Yes

5. Is the manuscript presented in an intelligible fashion and written in standard English?

Reviewer #1: Yes

Reviewer #2: Yes

6. Review Comments to the Author

Reviewer #1: This version of your paper is much improved comparing to the original one. As far I see this is proper research paper for publication.

Reviewer #2: (No Response)

7. PLOS authors have the option to publish the peer review history of their article (what does this mean?). If published, this will include your full peer review and any attached files.

Reviewer #1: **Yes: **Ahmadreza Asgharpourmasouleh

Reviewer #2: No

---

## [Editor Report · Acceptance letter]

15 Nov 2021

PONE-D-21-21013R1 

Unobtrusive Measures of Prejudice: Estimating percentages of public beliefs and behaviours. 

Dear Dr. Arnulf:

I'm pleased to inform you that your manuscript has been deemed suitable for publication in PLOS ONE. Congratulations! Your manuscript is now with our production department. 

Kind regards, 

on behalf of

Dr. Shah Md Atiqul Haq 

Academic Editor

PLOS ONE